# Prevalence and determinants of underweight, overweight, and obesity among reproductive-aged Bangladeshi women: Evidence from Bangladesh Demographic and Health Survey 2022

Ashim Kumar Nandi[ID][1][*], Kh Shafiur Rahaman[ID][2,3,4], Navira Chandio[2,3,5], Amit Arora[ID][2,3,5,6,7,8]

1 Department of Sociology, University of Barishal, Barishal, Bangladesh, 2 School of Medicine, Faculty of Health, Western Sydney University, Campbelltown, New South Wales, Australia, 3 Health Equity across Lifespan Research Laboratory, Campbelltown, New South Wales, Australia, 4 Bangladesh Academy of Dietetics and Nutrition (BADN), Bangladesh, 5 Translational Health Research Institute, Western Sydney University, Campbelltown Campus, Penrith, New South Wales, Australia, 6 Oral Health Services, Sydney Local Health District and Sydney Dental Hospital, NSW Health, Surry Hills, New South Wales, Australia, 7 Discipline of Child and Adolescent Health, Sydney Medical School, Faculty of Medicine and Health, The University of Sydney, Westmead, New South Wales, Australia, 8 ARCED Foundation, Bangladesh

☯ AKN and KSR are Joint First Authors.
* aknandi@bu.ac.bd

## Abstract

Over the past two decades, underweight, overweight, and obesity have increased in Bangladesh, becoming a major public health concern. This study investigated the current prevalence and examined the determinants of underweight, overweight, and obesity among reproductive-aged Bangladeshi women. Using data from a weighted sample of 9,213 women in the 2022 Bangladesh Demographic and Health Survey, we applied multinomial logistic regression to identify factors independently associated with being underweight, overweight, and obese. Among 9,213 reproductive-aged women, the prevalence of underweight, overweight, and obesity was 10.0%, 18.9%, and 36.5%, respectively. Multivariable analysis identified older age as the strongest predictor of overweight and obesity: women aged 35–49 years had nearly double the risk of overweight (ARRR = 2.01; 95% CI: 1.62–2.50) and more than triple the risk of obesity (ARRR = 3.56; 95% CI: 2.90–4.38) compared to those aged 15–24 years. Other significant factors included higher wealth status (ARRR = 1.94; 95% CI: 1.66–2.26 for obesity among wealthy households), media exposure (reading magazines: ARRR = 1.46; 95% CI: 1.14–1.87, watching television: ARRR = 1.28; 95% CI: 1.13–1.46), and parity (ARRR = 1.58; 95% CI: 1.23–2.02 for obesity among women with 1–4 children). Breastfeeding and agricultural/manual work protected against obesity. We also found that currently breastfeeding women (ARRR = 1.40; 95% CI: 1.13–1.72) and women who lived in Sylhet division (ARRR = 1.55; 95% CI: 1.13–2.12)

**Data availability statement:** Data associated with this study is available at https://dhsprogram.com/data/Using-Datasets-for-Analysis.cfm (accessed on 17 November 2024).

**Funding:** The author(s) received no specific funding for this work.

**Competing interests:** The authors have declared that no competing interests exist.

**Abbreviations:** AIC, Akaike Information Criterion; ARRR, Adjusted Relative Risk Ratio; BBS, Bangladesh Bureau of Statistics; BDHS, Bangladesh Demographic and Health Survey; BIC, Bayesian Information Criterion; BMI, Body Mass Index; BMRC, Bangladesh Medical Research Council; CI, Confidence Interval; EAs, Enumeration Areas; GDP, Gross Domestic Product; ICF, Inner City Fund; IMPS, Integrated Multi-Purpose Sampling; IRB, Institutional Review Board; IUD, Intra Uterine Device; LMICs, Low- and Middle-Income Countries; NIPORT, National Institute of Population Research and Training; PSU, Primary Sampling Unit; RRR, Relative Risk Ratio; SDM, Standard Days Method; SDGs, Sustainable Development Goals; URRR, Unadjusted Relative Risk Ratio; USAID, United States Agency for International Development.

were more likely to be underweight, while women with secondary or higher education, wealthier women, women aged 25 or higher, and women with 1–4 children were less likely to be underweight. Overall, a higher burden of underweight, overweight, and obesity exists in the sample. These findings emphasise the need for age- and context-specific interventions, such as promoting active lifestyles, regulating food environments, and incorporating nutrition education into maternal health programs, to combat rising obesity while addressing persistent underweight.

## Introduction

Underweight, overweight and obesity are linked to morbidities and increased mortality in adults [1]. Obesity is linked to several non-communicable diseases (NCDs) (e.g., cardiovascular disease, kidney disease, etc.) [2,3] and has become a public health concern because of increased prevalence in the past few decades [2]. According to a recent report, an estimated 1.11 billion adult women had obesity in 2021 [4]. Moreover, being overweight and obese can be connected with body dissatisfaction and diminished physical functioning, resulting in depression and social isolation [3]. Underweight is also linked to premature deaths [5], although the underweight prevalence has shown a decreasing trend over the past few years [6]. Nonetheless, a large proportion of people around the world continue to suffer from being underweight [7]. In women, being underweight, overweight, or obese leads to pregnancy-related complications (e.g., pre-eclampsia, eclampsia, gestational diabetes mellitus), infertility, and adverse birth outcomes such as abortion, pre-term birth, and neonatal and infant mortality [5,8]. Consequently, women are more vulnerable to adverse health outcomes than men [9].

Low- and middle-income countries (LMICs) are currently experiencing a double burden of nutrition with a greater underweight and overweight/obesity prevalence [10]. Bangladesh, an LMIC in South Asia with a population of over 160 million, is also subject to this double burden of nutrition [11]. The Bangladesh Demography and Health Survey [BDHS] (2011), a nationally representative survey, estimates that the prevalence of underweight, overweight, and obesity among adults was 30.4%, 18.9%, and 4.6%, respectively. The BDHS 2017–2018 reports that the prevalence of overweight and obesity surpasses underweight [11]. In both surveys, it was found that women were roughly twice as likely to be overweight/ obese compared to men. However, no such differences were observed in the case of underweight prevalence in both surveys [11,12]. For women, the prevalence of underweight was 15.49%, while the prevalence of overweight/ obesity was 45.60%, according to the BDHS 2017–2018 data [13]. In another study, more than half of the urban women were found to be overweight and obese [14]. Other South Asian countries have also reported a similar trend of malnutrition among reproductive-aged women, highlighting the urgent need to act on this nutritional health issue [15,16].

The determinants of underweight, overweight, and obesity could be environmental, behavioral, and individual [17]. Continued economic growth, growing

urbanization, predominantly in developing countries, globalization of food production, and alterations in dietary habits are the main contributing factors for increasing overweight and obesity [18]. Previous studies conducted among Bangladeshi adults reported that people with a low level of education and poor wealth status are more likely to be underweight. In contrast, people with a high educational background and substantial income are more prone to becoming overweight and obese [19–21]. A higher level of overweight and obesity has been reported in urban regions than in rural regions [22]. A high level of gender inequality in Bangladesh could be connected to the higher level of malnutrition among women than men [19].

In recent years, globalization, technological advancements, increased access to information through social media, the growing diversity of food production and choices, and rapid urbanization have significantly impacted our societies across the country. Hence, the lifestyle of individuals has also been affected, especially their eating habits and physical activity. However, previous studies have attempted to investigate the prevalence and determinants of underweight, overweight, and obesity among women and reported associated socio-economic factors [12–14,23,24]. Earlier studies have explored malnutrition among Bangladeshi women using BDHS 2017–2018 [11], substantial socioeconomic and lifestyle changes took place in recent years, including rapid urbanization, increased income levels (gross domestic product (GDP growth), and changes in the dietary patterns [25,26]. However, no recent nationally representative study has assessed these trends among reproductive-aged women using BDHS 2022 data.An updated analysis can also inform any changes in the relationships between underweight, overweight, obesity, and their determinants which is crucial for designing targeted interventions and informing national health policies aligned with Sustainable Development Goal 3 [27–29]. Therefore, our study aims to (i) estimate the prevalence of underweight, overweight, and obesity among reproductive-aged Bangladeshi women using BDHS 2022 data, and (ii) identify key socioeconomic and demographic factors associated with these conditions with the most recent data available in the country.

## Methods

### Data source

The 2022 Bangladesh Demographic and Health Survey (BDHS) was executed by the National Institute of Population Research and Training (NIPORT), the Medical Education and Family Welfare Division, the Ministry of Health and Family Welfare, and the Government of Bangladesh [30]. The Bangladesh Government and the United States Agency for International Development (USAID) provided financial aid [30]. Mitra and Associates surveyed from 27 June 2022 to 12 December 2022, while ICF supplied technical support under the DHS Program [30].

### Sampling design and sample size

The survey used the Integrated Multi-Purpose Sampling (IMPS) master sample, selected from a list of enumeration areas (EAs) for the 2011 Population and Housing Census provided by the Bangladesh Bureau of Statistics (BBS). The 2022 BDHS sample was stratified and selected in two stages [30]. At first, 675 EAs were chosen with a probability proportionate to their size and independent selection within each sampling stratum [30]. However, because of security concerns, the survey was ultimately executed in 674 clusters following the exclusion of one rural cluster in Cox's Bazar, Chattogram. Then, a specified number of 45 households per cluster was selected using an equal probability systematic choice from the newly generated household listing [30]. Out of 30,375 surveyed households, 30,053 ever-married women aged 15–49 years who were regular members of the selected families or who resided in the selected houses the night before they were interviewed, resulting in a response rate of 89.20%.

Following previous studies [31–34], this study excluded pregnant women or women who had given birth within the two months before the survey to reduce measurement bias associated with the initial weight increase from pregnancy and delivery. The study analyzed a weighted sample of 9,213 reproductive-aged women.

## Outcome measure

The outcome variable was the nutritional status of reproductive-age women, measured using the Lancet Diabetes and Endocrinology Commission's criteria [35] for the classification of Bangladeshi adult BMI. BMI is defined as the ratio of weight relative to height squared [30]. The survey measured weight using SECA scales with a digital display (model number SECA 874U) and measured height and length using a ShorrBoard® measuring board [30]. BMI was categorised into four groups, which subsequently constituted the study outcome variables:

• Underweight: BMI < 18.50 kg/m$^2$

• Normal weight: BMI >= 18.50 kg/m$^2$ and BMI < 23.00 kg/m$^2$

• Overweight: BMI >= 23.00 kg/m$^2$ and BMI < 25.00 kg/m$^2$

• Obesity: BMI >= 25.00 kg/m$^2$

## Independent variables

Independent variables were chosen based on prior research from LMICs [31–33,35–42] and data availability in the BDHS. These variables were categorized into socio-economic, demographic, behavioral, and regional factors.

Socio-economic factors included women's education (categorized as 'no education,' 'primary,' or 'secondary or higher'. Please note that during the data collection, women's education included 'no education,' 'primary,' 'secondary,' and 'higher' categories. For the analysis, the current study recoded the 'secondary' and 'higher' categories into 'secondary or higher'.), women's employment (categorized as 'not working,' 'professional/semi-professional,' or 'agricultural/manual'. Please note that the 'professional/semi-professional' category included 'professional/technical/managerial,' 'sales,' 'agricultural—self-employed,' and 'skilled manual' categories. The 'agricultural/manual' category incorporated 'household and domestic,' and 'unskilled manual' categories.), household wealth (categorized as 'poor,' 'middle,' or 'rich'. Please note that the 'poor' category included the 'poorest' and 'poorer' categories, while the 'rich' category included 'richer' and 'richest' categories.), while demographic and behavioral factors included marital status (categorized as 'currently married,' or 'formerly married'. Please note that the 'formerly married' category included 'widowed,' 'divorced,' and 'no longer living together/separated' categories.), women's age (categorized as '15-24,' '25-34,' or '35-49'), parity (categorized as '0,' '1-4,' or '5 or more'. Please note that the '5 or more' category included a total of 5–11 children.), listening to the radio (categorized as 'no,' or 'yes'. Please note that the 'yes' category included, 'less than once a week', or 'at least once a week' categories.), reading a magazine (categorized as 'no,' or 'yes'. Please note that the 'yes' category included respondents who answered, 'less than once a week', or 'at least once a week'.), watching television (categorized as 'no,' or 'yes'. Please note that the 'yes' category included respondents who answered, 'less than once a week', or 'at least once a week'.), contraceptive use (categorized as 'not using,' or 'using'. Please note that the 'using' category included 'pill,' 'IUD,' 'injections,' 'male condom,' 'female sterilization', 'male sterilization', 'periodic abstinence', 'withdrawal', 'other traditional', 'implants/Norplant,' 'lactational amenorrhea [lam]', 'female condom', 'emergency contraception', 'other modern method', and 'standard days method [SDM]'.), and currently breastfeeding (categorized as 'no,' or 'yes').

Regional factors incorporated place of residence (categories are 'urban' and 'rural'), and division (i.e., region of residence) (categories are 'Barishal,' 'Chattogram,' 'Dhaka,' 'Khulna,' 'Mymensingh,' 'Rajshahi,' 'Rangpur,' and 'Sylhet').

## Statistical analysis

The descriptive features of studied individuals were analysed by calculating weighted counts, which involved calculating percentages for each research variable. The next phase assessed the prevalence of underweight, overweight, and obesity according to socioeconomic, demographic, behavioural, and community-level factors.

Predictor variables that were found to be significant ($p < 0.05$) in the bivariate analysis were considered for multivariable analysis. Survey-weighted multinomial logistic regression modelling was employed to investigate the relationship between individual-level characteristics (socioeconomic, demographic, and behavioural) and community-level factors with (a) underweight, (b) overweight, and (c) obesity, using normal weight groups as the reference category. A previous study [31] pointed out that multilevel modelling has advantages over single-level logistic regression models for the following reasons: (1) multilevel modelling considers the hierarchical nature of data, ensuring that statistical significance is not overstated [35,42], (2) it considers the dependence of observations in the same clusters and allows for simultaneous estimation of cluster-level effects and community-level predictors, preventing overstatement of statistical significance [43].

The current study specified regression models in four stages, like earlier studies (e.g., [31]). At first, we constructed a null unconditional model without research variables. The second step incorporated individual-level characteristics (socio-economic, demographic, and behavioural) into the model. In stage three, regional parameters, precisely the place of residence and division, were incorporated without the variables from stage two. The final model incorporated individual and regional characteristics—the results of the final model are presented in the results section (see Table 2).

Relative risk ratios (RRR) and their accompanying 95% confidence intervals (CIs) were estimated to assess the connection between the study parameters and dependent variables. Like previous studies [31,44], the study employed the Pseudo-$R^2$, log-likelihood, Akaike information criterion (AIC), and Bayesian information criterion (BIC) to test the goodness of fit (see S1 Table). The Variance Inflation Factor (VIF) was employed to assess multicollinearity, which was assessed using the 'estat vif' command. A VIF greater than 10 indicates a high collinearity [45]. The VIF values in our study ranged from 1.08 to 2.90 (see S2 Table). All statistical analyses were performed using Stata version 18.0 (StataCorp, USA), using the 'svyset' and 'svy' commands to account for sample weights, clustering effects, and stratification. The 'mlogit' command was utilized for multinomial models.

### Ethical considerations

The Institutional Review Board (IRB) of ICF International evaluates and authorizes the methods and questions for standard DHS surveys. Alongside the data obtained from interviews, biomarker data (e.g., height and weight) were gathered from a subsample of households in the 2022 BDHS [30]. ICF collaborated with local specialists to develop the biomarker testing procedure, obtaining permission from the ICF Institutional Review Board and the Bangladesh Medical Research Council (BMRC) [30]. The Biomarker Questionnaire was completed on paper and input into a computer-assisted personal interviewing (CAPI) system before the team left the cluster [30]. The data files contain no individual names or residential addresses [30]. The geographic identifiers are limited to the regional level, where regions often consist of extensive geographical areas. Every enumeration region (Primary Sampling Unit) is assigned a PSU number in the data file; however, these PSU numbers lack labels to denote their names or locations [30]. So, authors lacked access to information that could identify individual participants during or subsequent data collection. All respondents of the survey provided written informed consent. On 16 September 2024, we received written permission from ICF and the Demographic and Health Survey (DHS) Program to use the Bangladesh DHS datasets for this study. We last accessed the 2022 BDHS datasets on 10 April 2025 for our study purposes.

### Results

### Characteristics of the study participants

Among the 9,213 ever-married women of reproductive age, aged 15–49 years, analyzed, 10.0% were classified as underweight, 34.6% as normal weight, 18.9% as overweight, and 36.5% as obese according to the Asian cut-off for BMI. As part of the sensitivity analysis, the prevalence of BMI according to the global cut-off for BMI was also estimated, which is presented in Fig 1. The difference in prevalence rate due to changes in BMI cut-off shows that more women are at health risks in Bangladesh due to obesity, and fewer women are in normal weight than we found using the global cut-off.

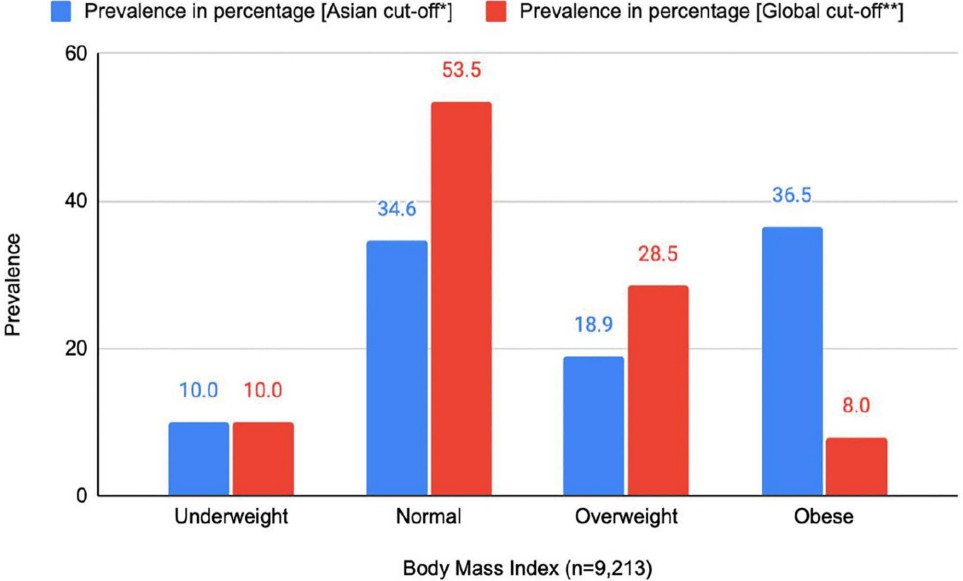

**Fig 1. Prevalence of BMI categories among reproductive-aged women (15-49 years) in Bangladesh (2022).** Note: * As discussed in the above-mentioned "Methods" section: Underweight: BMI < 18.50 kg/m²; Normal weight: BMI >=18.50 kg/m² and BMI < 23.00 kg/m²; Overweight: BMI >=23.00 kg/m² and BMI < 25.00 kg/m²; Obesity: BMI >=25.00 kg/m² ** WHO classification for BMI: Underweight: BMI < 18.50 kg/m²; Normal weight: BMI >=18.50 kg/m² and BMI < 25.00 kg/m²; Overweight: BMI >=25.00 kg/m² and BMI < 30.00 kg/m²; Obesity: BMI >=30.00 kg/m² [26].

Among the women, 14.2% had no education, 62.7% were not working, 37.3% came from poor households and 71.4% lived in rural areas. Of the women, 8.4% had no children, 22.4% were aged between 15 and 24, and 20.7% were breast-feeding. 2.7% of the women were listening to radio, 6.7% were reading magazines, while 55.7% were watching television (see Table 1).

## Prevalence of underweight, overweight, and obesity among reproductive-age women in Bangladesh

The prevalence of underweight was higher among reproductive-age women who had no education (13.3%) compared to women who attained primary (10.1%) or secondary or higher education (9.2%). The youngest women (i.e., age 15–24) (17.8%) had a higher prevalence of underweight than the older women (i.e., age 35–49) (7.5%), while the youngest women (15.2%) had a lower prevalence of overweight than older women (20.6%). Women who resided in wealthy households (47.0%) had a higher prevalence of obesity than women who resided in poor households (25.9%). In comparison, women who resided in poor households (15.5%) had a higher underweight prevalence than those in rich households (5.7%). Breastfeeding women (16.2%) had a higher prevalence of being underweight than non-breastfeeding women (8.4%) while breastfeeding women had a lower prevalence of obesity (24.1%) than non-breastfeeding women (39.7%). Women who watched television (41.2%) had a higher prevalence of obesity compared to those who had not watched television (30.5%). Rural women (11.3%) had a higher prevalence of underweight than women in urban areas (6.6%). In contrast, urban women (45.5%) had a higher obesity prevalence than rural women (32.9%) (see Table 1).

## Univariable analysis

Women with secondary or higher education, women whose employment was professional/semi-professional, women who resided in middle-income and wealthy households, women who belonged to age groups 25–34 and 35–49, women with 1–4 children, and women who read a magazine were at a lower risk of falling into an underweight group. In contrast,

**Table 1. Distribution and the prevalence of underweight, normal weight, overweight, and obesity by socioeconomic, demographic, behavioral, and regional characteristics in Bangladesh, BDHS 2022 (n=9,213, except where indicated).**

| Variables | ᵃN (%) | Underweight | Normal weight | Overweight | Obesity |
|---|---|---|---|---|---|
|  |  | ᵃn (%) | n (%) | n (%) | n (%) |
| *Socio-economic factors* |  |  |  |  |  |
| **Women's education\*\*\*** |  |  |  |  |  |
| No education | 1309 (14.2) | 174 (13.3) | 494 (37.7) | 273 (20.9) | 368 (28.1) |
| Primary | 2438 (26.5) | 247 (10.1) | 868 (35.6) | 453 (18.6) | 871 (35.7) |
| Secondary or higher | 5467 (59.3) | 502 (9.2) | 1828 (33.4) | 1014 (18.6) | 2122 (38.8) |
| **Women's employment \*\*\* (n=8906)** |  |  |  |  |  |
| Not working | 5588 (62.7) | 571 (10.2) | 1924 (34.4) | 1003 (18.0) | 2090 (37.4) |
| Professional/semi-professional | 705 (7.9) | 41 (5.8) | 210 (29.9) | 146 (20.7) | 307 (43.6) |
| Agricultural/manual | 2613 (29.4) | 270 (10.3) | 922 (35.3) | 542 (20.7) | 880 (33.7) |
| **Household wealth\*\*\*** |  |  |  |  |  |
| Poor | 3440 (37.3) | 533 (15.5) | 1407 (40.9) | 608 (17.7) | 892 (25.9) |
| Middle | 1927 (20.9) | 173 (9.0) | 716 (37.2) | 379 (19.6) | 660 (34.2) |
| Rich | 3846 (41.8) | 217 (5.7) | 1066 (27.7) | 754 (19.6) | 1809 (47.0) |
| *Demographic and behavioral factors* **Marital status** |  |  |  |  |  |
| Currently married | 8738 (94.8) | 865 (9.9) | 3027 (34.6) | 1649 (18.9) | 3198 (36.6) |
| Formerly married | 475 (5.2) | 58 (12.1) | 163 (34.3) | 91 (19.2) | 164 (34.4) |
| **Women's age\*\*\*** |  |  |  |  |  |
| 15-24 | 2067 (22.4) | 366 (17.8) | 982 (47.5) | 315 (15.2) | 404 (19.5) |
| 25-34 | 3153 (34.3) | 258 (8.2) | 1041 (33.0) | 601 (19.0) | 1253 (39.8) |
| 35-49 | 3993 (43.3) | 299 (7.5) | 1166 (29.2) | 824 (20.6) | 1704 (42.7) |
| **Parity\*\*\*** |  |  |  |  |  |
| 0 | 776 (8.4) | 125 (16.0) | 339 (43.7) | 116 (14.9) | 197 (25.4) |
| 1-4 | 7864 (85.4) | 720 (9.2) | 2635 (33.5) | 1508 (19.2) | 3000 (38.1) |
| 5 or more | 574 (6.2) | 78 (13.6) | 215 (37.5) | 116 (20.3) | 164 (28.6) |
| **Listening to radio** |  |  |  |  |  |
| No | 8960 (97.3) | 902 (10.1) | 3101 (34.6) | 1702 (19.0) | 3256 (36.3) |
| Yes | 253 (2.7) | 21 (8.4) | 89 (35.0) | 38 (15.0) | 105 (41.6) |
| **Reading magazine\*\*\*** |  |  |  |  |  |
| No | 8591 (93.3) | 891 (10.4) | 3019 (35.1) | 1641 (19.1) | 3041 (35.4) |
| Yes | 622 (6.7) | 32 (5.1) | 170 (27.4) | 99 (16.0) | 320 (51.5) |
| **Watching television\*\*\*** |  |  |  |  |  |
| No | 4081 (44.3) | 479 (11.7) | 1577 (38.6) | 779 (19.1) | 1246 (30.5 |
| Yes | 5132 (55.7) | 444 (8.7) | 1613 (31.4) | 961 (18.7) | 2115 (41.2) |
| **Currently breastfeeding\*\*\*** |  |  |  |  |  |
| No | 14827 (79.3) | 615 (8.4) | 2359 (32.3) | 1433 (19.6) | 2903 (39.7) |
| Yes | 1903 (20.7) | 307 (16.2) | 831 (43.6) | 307 (16.1) | 459 (24.1) |
| **Contraceptive use** |  |  |  |  |  |
| Not using | 3197 (34.7) | 320 (10.0) | 1056 (33.0) | 592 (18.5) | 1229 (38.5) |
| Using | 6016 (65.3) | 603 (10.0) | 2134 (35.5) | 1148 (19.1) | 2132 (35.4) |
| *Regional factors* **Place of residence\*\*\*** |  |  |  |  |  |
| Urban | 2640 (28.6) | 175 (6.6) | 775 (29.4) | 488 (18.5) | 1202 (45.5) |
| Rural | 6574 (71.4) | 747 (11.3) | 2415 (36.7) | 1252 (19.1) | 2159 (32.9) |

*(Continued)*

**Table 1.** (Continued)

| Variables | ªN (%) | Underweight | Normal weight | Overweight | Obesity |
|---|---|---|---|---|---|
| | | ªn (%) | n (%) | n (%) | n (%) |
| **Division*** | | | | | |
| Barishal | 554 (6.0) | 59 (10.6) | 190 (34.4) | 101 (18.2) | 204 (36.8) |
| Chattogram | 1671 (18.1) | 140 (8.4) | 533 (31.9) | 325 (19.5) | 672 (40.2) |
| Dhaka | 2322 (25.2) | 176 (7.6) | 761 (32.7) | 434 (18.7) | 952 41.0) |
| Khulna | 1087 (11.8) | 80 (7.4) | 362 (33.4) | 213 (19.6) | 432 (39.6) |
| Mymensingh | 719 (7.8) | 118 (16.5) | 301 (41.8) | 119 (16.6) | 181 (25.1) |
| Rajshahi | 1246 (13.5) | 132 (10.6) | 415 (33.3) | 245 (19.6) | 455 (36.5) |
| Rangpur | 1077 (11.7) | 132 (12.3) | 416 (38.6) | 209 (19.4) | 320 (29.7) |
| Sylhet | 537 (5.9) | 86 (16.1) | 211 (39.3) | 94 (17.5) | 146 (27.1) |

ªN and n indicate the weighted count and % indicates the column percentage.

*** $p < 0.001$ (P value indicates $x^2$ test results in bivariate analysis).

breastfeeding women, women who lived in rural areas, and women who lived in Mymensingh, Rajshahi, Rangpur, and Sylhet divisions were at a higher risk of falling into the underweight group. While currently breastfeeding women, rural women, and women who lived in the Mymensingh division were at a lower risk of falling into overweight group, women who were professional/semi-professional, women who resided in middle and wealthy households, women who belonged to age groups 25–34 and 35–49, women with 1 or more children, and women who watched television were at a higher risk of falling into overweight group. Moreover, women who worked in agricultural or manual sectors, currently breastfeeding women, women who were using contraceptives, women who lived in rural areas and in the Mymensingh, Rangpur, and Sylhet divisions were at a lower risk being obese. In comparison, women with primary, secondary or higher education, women whose employment was professional/semi-professional, women who resided in middle-income and wealthy households, women who belonged to age groups 25–34 and 35–49, women with 1–4 children, women who read magazines, and women who watched television were at were at a higher risk being obese. (see Table 2).

## Multivariable analysis

**Factors associated with underweight in Bangladeshi women.** Reproductive-aged women who resided in middle (RRR = 0.66; 95% confidence interval [CI]: 0.53, 0.83) or rich households (RRR = 0.59; 95% CI: 0.47, 0.74) were less likely to be underweight compared to women who resided in poor households. Women with secondary or higher levels of education were less likely to be underweight, compared to women with no education (RRR = 0.59; 95% CI: 0.47, 0.74). Compared to women aged between 15 and 24 years, older women were less likely to be underweight (age group 25–34: RRR = 0.67; 95% CI: 0.54, 0.84 and age group 35–49: RRR = 0.67; 95% CI: 0.52, 0.87). Mothers of one to four children were less likely to be underweight compared to non-mothers (RRR = 0.73; 95% CI: 0.55, 0.98). Breastfeeding women were more likely to be underweight than non-breastfeeding women (RRR = 1.40; 95% CI: 1.13, 1.72). Women who resided in the Sylhet division were more likely to be underweight compared to women who lived in the Dhaka division (RRR = 1.55; 95% CI: 1.13, 2.12) (see Table 3).

**Factors associated with overweight in Bangladeshi women.** Women from wealthy households (RRR = 1.53; 95% CI: 1.28, 1.82) (compared to poor households), those in the age groups of 25–34 (RRR = 1.63; 95% CI: 1.34, 1.98) and 35–49 (RRR = 2.01; 95% CI: 1.62, 2.50) (compared to 15–24), and those with 1–4 children (RRR = 1.52; 95% CI: 1.14, 2.02) (compared to women with no children) are at a higher risk of overweight. Conversely, breastfeeding women are at a lower risk of being overweight than non-breastfeeding women (RRR = 0.71; 95% CI: 0.59, 0.86) (see Table 3).

**Table 2. Unadjusted Relative Risk Ratio (URRR) of underweight, overweight, and obesity by socioeconomic, demographic, behavioral, and regional factors in Bangladesh.**

| Variables | Underweight | Overweight | Obesity |
|---|---|---|---|
| | [a]URRR (95% CI[b]) | URRR (95% CI) | URRR (95% CI) |
| *Socio-economic factors* Women's education | | | |
| No education | 1.00 | 1.00 | 1.00 |
| Primary | 0.81 (0.64, 1.03) | 0.94 (0.77, 1.16) | 1.34 (1.12, 1.61) |
| Secondary or higher | 0.78 (0.63, 0.97) | 1.00 (0.84, 1.20) | 1.56 (1.32, 1.83) |
| **Women's employment** | | | |
| Not working | 1.00 | 1.00 | 1.00 |
| Professional/semi-professional | 0.65 (0.43, 0.98) | 1.33 (1.02, 1.73) | 1.34 (1.08, 1.67) |
| Agricultural/manual | 0.99 (0.82, 1.18) | 1.13 (0.98, 1.30) | 0.88 (0.78, 0.99) |
| **Household wealth** | | | |
| Poor | 1.00 | 1.00 | 1.00 |
| Middle | 0.64 (0.52, 0.79) | 1.22 (1.03, 1.46) | 1.45 (1.25, 1.46) |
| Rich | 0.54 (0.44, 0.65) | 1.64 (1.41, 1.90) | 2.68 (2.36, 3.04) |
| *Demographic and behavioral factors* Marital status | | | |
| Currently married | 1.00 | 1.00 | 1.00 |
| Formerly married | 1.24 (0.90, 1.70) | 1.03 (0.77, 1.38) | 0.95 (0.74, 1.22) |
| **Women's age** | | | |
| 15-24 | 1.00 | 1.00 | 1.00 |
| 25-34 | 0.66 (0.54, 0.81) | 1.80 (1.50, 2.16) | 2.93 (2.49, 3.44) |
| 35-49 | 0.69 (0.57, 0.83) | 2.20 (1.85, 2.63) | 3.55 (3.04, 4.15) |
| **Parity** | | | |
| 0 | 1.00 | 1.00 | 1.00 |
| 1-4 | 0.74 (0.58, 0.95) | 1.68 (1.30, 2.17) | 1.96 (1.59, 2.41) |
| 5 or more | 0.99 (0.70, 1.41) | 1.58 (1.12, 2.24) | 1.31 (0.97, 1.77) |
| **Listening to radio** | | | |
| No | 1.00 | 1.00 | 1.00 |
| Yes | 0.82 (0.48, 1.42) | 0.78 (0.50, 1.21) | 1.13 (0.82, 1.56) |
| **Reading magazine** | | | |
| No | 1.00 | 1.00 | 1.00 |
| Yes | 0.64 (0.42, 0.98) | 1.07 (0.81, 1.42) | 1.87 (1.50, 2.32) |
| **Watching television** | | | |
| No | 1.00 | 1.00 | 1.00 |
| Yes | 0.91 (0.77, 1.06) | 1.21 (1.06, 1.37) | 1.66 (1.49, 1.85) |
| **Currently breastfeeding** | | | |
| No | 1.00 | 1.00 | 1.00 |
| Yes | 1.42 (1.19, 1.69) | 0.61 (0.52, 0.72) | 0.45 (0.39, 0.52) |
| **Contraceptive use** | | | |
| Not using | 1.00 | 1.00 | 1.00 |
| Using | 0.93 (0.79, 1.10) | 0.96 (0.83, 1.10) | 0.86 (0.77, 0.96) |
| *Regional factors* Place of residence | | | |
| Urban | 1.00 | 1.00 | 1.00 |
| Rural | 1.37 (1.12, 1.67) | 0.82 (0.71, 0.96) | 0.58 (0.51, 0.65) |
| **Division** | | | |
| Dhaka | 1.00 | 1.00 | 1.00 |
| Barishal | 1.33 (0.96, 1.84) | 0.93 (0.72, 1.20) | 0.86 (0.70, 1.05) |

*(Continued)*

**Table 2.** (Continued)

| Variables | Underweight | Overweight | Obesity |
|---|---|---|---|
| | [a]URRR (95% CI[b]) | URRR (95% CI) | URRR (95% CI) |
| Chattogram | 1.14 (0.83, 1.56) | 1.07 (0.85, 1.35) | 1.01 (0.84, 1.22) |
| Khulna | 0.96 (0.69, 1.32) | 1.03 (0.82, 1.30) | 0.95 (0.79, 1.15) |
| Mymensingh | 1.70 (1.28, 2.27) | 0.69 (0.54, 0.88) | 0.48 (0.39, 0.59) |
| Rajshahi | 1.38 (1.01, 1.87) | 1.03 (0.82, 1.31) | 0.88 (0.72, 1.07) |
| Rangpur | 1.37 (1.02, 1.85) | 0.88 (0.70, 1.11) | 0.62 (0.51, 0.75) |
| Sylhet | 1.77 (1.32, 2.37) | 0.78 (0.61, 1.00) | 0.55 (0.45, 0.68) |

[a] URRR: Unadjusted relative risk ratio.

[b] 95%CI: 95% confidence interval.

**Factors associated with obesity in Bangladeshi women.** Women with primary (RRR = 1.39; 95% CI: 1.14, 1.70), secondary or higher education (RRR = 1.57; 95% CI: 1.28, 1.92) (compared to women with no education), women who resided in middle (RRR = 1.23; 95% CI: 1.04, 1.44) or wealthy households (RRR = 1.94; 95% CI: 1.66, 2.26) (compared to poor households), women who were belong to age group 25–34 (RRR = 2.79; 95% CI: 2.32, 3.36) and 35–49 (RRR = 3.56; 95% CI: 2.90, 4.38) (compared to age group 15–24), women with 1–4 numbers of children (RRR = 1.58; 95% CI: 1.23, 2.02) (compared to women with no children), women who read magazine (RRR = 1.46; 95% CI: 1.14, 1.87) (compared to women who had not read magazine), and women who watched television (RRR = 1.28; 95% CI: 1.13, 1.46) (compared to women who had not watched television) were at a higher risk of being obese (compared to being normal weight). Women who had lived in Chattogram division (RRR = 1.36; 95% CI: 1.12, 1.67) were more likely to be obese compared to women who lived in Dhaka division, while women who resided in Mymensingh (RRR = 0.69; 95% CI: 0.55, 0.86) and Sylhet (RRR = 0.69; 95% CI: 0.55, 0.87) were less likely to be obese compared to women living in Dhaka. Moreover, breastfeeding women (RRR = 0.56; 95% CI: 0.47, 0.66) (compared to non-breastfeeding women) and women who had worked in agricultural/manual sectors (RRR = 0.79; 95% CI: 0.69, 0.91) (compared to women who had no employment) were at a lower risk of being obese (compared to the risk of being normal weight) (see Table 3).

## Discussion

Our study examined the prevalence and determinants of underweight, overweight, and obesity among reproductive-aged women in Bangladesh, and included a range of demographic and socio-economic variables such as parity, media engagements, contraceptive use, and breastfeeding status compared to the previous studies [19,46]. Analysing the nationally representative data, we found a double burden of malnutrition among reproductive-aged Bangladeshi women. Underweight is clustered among younger, less-educated, poorer, rural, and currently breastfeeding women. In contrast, overweight and obesity were concentrated in older age-groups, educated (primary and above), wealthier, urban women, and those with higher media exposure (television/ magazine). In terms of geographical distribution, obesity risk was higher in Chattogram and lower in Mymensingh and Sylhet compared with the capital (Dhaka). Engaging in agricultural/manual labour and breastfeeding helped to reduce the risk of obesity, whereas holding professional or semi-professional jobs and having multiple children were associated with a higher risk [31].

In our study, 10% of women were found to be underweight. An earlier study has reported slightly higher rates of underweight in females (15.49%) using the BDHS 2017–2018 data [13]. Even though the rate of underweight individuals has decreased in Bangladesh, especially among children and women, the issue of overweight and obesity is rising daily [47,48]. In our study population, 18.9% and 36.5% of women of reproductive age were found to be overweight and obese. This is significantly higher than the previous year reported by Gupta and Kibria (2021) (14.1%) [11]. In 2021, the prevalence of overweight and obesity was 39.4%, 47.8%, and 68.7% among adult females in India, Nepal, and Pakistan,

**Table 3. Adjusted Relative Risk Ratio (ARRR) of underweight, overweight, and obesity by socioeconomic, demographic, behavioral, and regional factors in Bangladesh (n = 8,903).**

| Variables[a] | Underweight | Overweight | Obesity |
|---|---|---|---|
| | [b]ARRR (95% CI[c]) | ARRR (95% CI[#]) | ARRR (95% CI[#]) |
| *Socioeconomic factors* Women's education | | | |
| No education | 1 | 1 | 1 |
| Primary | 0.76 (0.58, 1.00) | 1.00 (0.80, 1.25) | 1.39 (1.14, 1.70) |
| Secondary or higher | 0.76 (0.58, 0.99) | 1.14 (0.91, 1.43) | 1.57 (1.28, 1.92) |
| **Women's employment** | | | |
| Not working | 1 | 1 | 1 |
| Professional/semi-professional | 0.85 (0.56, 1.29) | 1.10 (0.84, 1.44) | 0.90 (0.71, 1.14) |
| Agricultural/manual | 1.08 (0.89, 1.31) | 1.02 (0.87, 1.18) | 0.79 (0.69, 0.91) |
| **Household wealth** | | | |
| Poor | 1 | 1 | 1 |
| Middle | 0.66 (0.53, 0.83) | 1.14 (0.94, 1.37) | 1.23 (1.04, 1.44) |
| Rich | 0.59 (0.47, 0.74) | 1.53 (1.28, 1.82) | 1.94 (1.66, 2.26) |
| *Demographic and behavioral factors* Women's age | | | |
| 15-24 | 1 | 1 | 1 |
| 25-34 | 0.67 (0.54, 0.84) | 1.63 (1.34, 1.98) | 2.79 (2.32, 3.36) |
| 35-49 | 0.67 (0.52, 0.87) | 2.01 (1.62, 2.50) | 3.56 (2.90, 4.38) |
| **Parity** | | | |
| 0 | 1 | 1 | 1 |
| 1-4 | 0.73 (0.55, 0.98) | 1.52 (1.14, 2.02) | 1.58 (1.23, 2.02) |
| 5 or more | 0.92 (0.60, 1.42) | 1.29 (0.86, 1.92) | 1.07 (0.75, 1.52) |
| **Reading magazine** | | | |
| No | 1 | 1 | 1 |
| Yes | 0.73 (0.47, 1.13) | 0.95 (0.70, 1.29) | 1.46 (1.14, 1.87) |
| **Watching television** | | | |
| No | 1 | 1 | 1 |
| Yes | 1.07 (0.90, 1.28) | 1.08 (0.94, 1.26) | 1.28 (1.13, 1.46) |
| **Currently breastfeeding** | | | |
| No | 1 | 1 | 1 |
| Yes | 1.40 (1.13, 1.72) | 0.71 (0.59, 0.86) | 0.56 (0.47, 0.66) |
| *Regional factors* Place of residence | | | |
| Urban | 1 | 1 | 1 |
| Rural | 1.07 (0.86, 1.34) | 0.97 (0.82, 1.14) | 0.82 (0.72, 0.94) |
| **Division** | | | |
| Dhaka | 1 | 1 | 1 |
| Barishal | 1.07 (0.75, 1.51) | 1.08 (0.83, 1.41) | 1.22 (0.98, 1.53) |
| Chattogram | 0.99 (0.71, 1.39) | 1.22 (0.96, 1.55) | 1.36 (1.12, 1.67) |
| Khulna | 0.88 (0.62, 1.23) | 1.05 (0.83, 1.34) | 1.12 (0.92, 1.38) |
| Mymensingh | 1.35 (0.99, 1.83) | 0.80 (0.62, 1.03) | 0.69 (0.55, 0.86) |
| Rajshahi | 1.20 (0.87, 1.68) | 1.06 (0.83, 1.35) | 1.05 (0.85, 1.29) |
| Rangpur | 1.11 (0.81, 1.54) | 1.02 (0.80, 1.30) | 0.94 (0.76, 1.17) |
| Sylhet | 1.55 (1.13, 2.12) | 0.86 (0.66, 1.12) | 0.69 (0.55, 0.87) |

[a]All variables in the final model were variables for which, when excluded, the change in deviance compared with the corresponding $\chi^2$ test statistic on the relevant degrees of freedom was significant.

[b]ARRR: Adjusted relative risk ratio.

[c]95%CI: 95% confidence interval.

respectively [4]. This discrepancy can likely be attributed to variations in categorization criteria for body weight and differences in selected sociodemographic factors. Studies from other South Asian countries have also suggested a similar rise in obesity rates [14]. However, overweight and obesity are becoming increasingly prevalent among urban women in Bangladesh [14]. This increase is attributed mainly to shifting socioeconomic conditions, evolving lifestyles, and changing dietary habits [49]. The growing prevalence of overweight and obesity in Bangladesh highlights an rising risk of NCDs among its population, particularly women [6].

The prevalence of overweight and obesity was found to be the highest among women between the ages of 25–34 years and 35–49 years [14]. Another study conducted in Bangladesh also reported a similar finding [13]. This aligns with studies from India [50] and other Asian countries [16], where older women tend to have lower physical activity levels and are more prone to consuming calorie-dense diets. Additionally, age-related changes in body composition make women over 30 more susceptible to weight gain than younger age groups [51]. Moreover, women aged below 25 years are more likely to be categorized as underweight compared to older women [23]. This is similar to what we found in our study, where younger women between the ages of 15–24 years are being categorised as underweight. Studies indicate that Bangladeshi women have higher rates of physical inactivity than women in other South Asian nations [14]. The other reason is that young adolescent Bangladeshi women are becoming more cautious about their health. Other research carried out in Bangladesh and India claims that there is a high prevalence of underweight women in rural areas [23,49,50]. This current study also found that the percentage of obesity among reproductive-aged women is higher among urban residents compared to rural residents. This highlights the differences in socioeconomic and lifestyle factors between the two settings.

Education and financial status play a crucial role in shaping women's nutritional outcomes. Few studies in Bangladesh have found a clear link between these factors and a woman's likelihood of being underweight or overweight. Generally, as education and wealth increase, the risk of undernutrition decreases [23,49]. In our study, having no education or lower education was associated with being underweight. An earlier study has also reported that being underweight was negatively correlated with wealth index and higher educational attainment [13]. However, an inverse trend is observed with overweight issues—higher education and wealth often correlate with a greater risk of obesity. In many LMICs, being overweight and obese are often perceived as indicators of higher socioeconomic status [51–53]. Previous studies have also reported that higher education and greater household wealth were strongly linked to overweight and obesity in Bangladesh [11,14]. This pattern has also been observed in other LMICs [1,54–56]. Consistent with previous research, overweight and obesity were linked with socioeconomic status, whereas higher education levels were only linked with obesity in our study. Women living in wealthier households had a higher risk of being overweight and obese. Wealthier, well-educated individuals in LMICs tend to lead more sedentary lifestyles and consume energy-dense, nutrient-poor foods, and lack of physical exercise, increasing their risk of obesity [57,58]. As a result, socioeconomic determinants such as income, education, and wealth contribute significantly to the increasing obesity epidemic, subsequently driving up NCD rates [59–61].

A study in Bangladesh identifies a strong association between the number of pregnancies and overweight/obesity [14]. Older women and those with multiple pregnancies were more likely to be overweight or obese, whereas they had a lower likelihood of being underweight [14]. In our study, we found that women with several pregnancies (between 1–4 times) had a higher risk of being overweight and obese. Potential contributing factors include the hormonal changes linked to pregnancy that may promote weight gain [62]. The use of hormonal contraceptives, which have been speculated to cause weight gain, though this study did not find a significant association between contraceptive use and overweight/ obesity. A similar finding was observed in a study in Bangladesh [14]. A recent systematic review also found insufficient evidence to confirm this link [63].

Our study found no significant link between women's marital status and underweight, overweight, or obesity. For women, being married, separated, divorced, or widowed was associated with lower odds of being underweight, a pattern reported by many previous studies [11,13,64]. In our adjusted results, agricultural or manual work appears to be protective

against obesity compared to non-employment or professional roles. National multi-level analysis of BDHS 2017–2018 found lower odds of overweight/ obesity among working women and higher risk among those in rich communities, highlighting the role of physical activity in occupational and community settings [65]. But in professional women, extended periods of sitting and a diet high in calories are linked to overweight and obesity [39].

Our study also found that women in rural Bangladesh had a significantly higher risk of being underweight compared to their counterparts. This disparity stems from the dual burden of malnutrition, where poverty leads to chronic undernutrition while urban areas experience rising rates of obesity [23,66]. From a public health perspective, this issue demands urgent intervention through educational awareness programs. Fast urbanization and ongoing economic development in many LMICs are directly linked to increasing obesity rates. Several factors contribute to this trend, including sedentary lifestyles, reliance on motorized transport, reduced physical activity, and a growing preference for fast and processed foods [52,67]. The shift toward modern technology, which minimizes the need for physical effort, further exacerbates the issue [23]. However, our study did not find any link between obesity and place of residence among women. Another study conducted among adults in Bangladesh in recent years also did not find any association between residential location and nutritional status [11].

Compared to Dhaka, women living in the Chattogram division had a higher risk of obesity, and women from the Sylhet division had a higher risk of being underweight in this study. Another study also reported that adult participants from Sylhet had lower odds of being overweight/obese compared to Dhaka residents [11]. Such differences, previously reported in other studies (Biswas et al., 2017), warrant further investigation [12].

Our study reported that watching television is one of the associated factors for obesity. This is in line with the study that took place in Bangladesh and Myanmar, where the prevalence of overweight and obesity was linked to watching television at least once a week [68–69]. A similar correlation has been documented in the high-income nations, such as the United States and Australia [70–73]. Research reported that viewing television can result in unhealthy food choices because of the continued promotion of sugary drinks [74–75]. Moreover, a rise in junk food consumption is typically linked to elevated socioeconomic status, and having a television at home serves as an indicator of individuals' higher socioeconomic status in LMICs [31]. Our study also reported that women who were breastfeeding had a higher chance of being underweight. A study in India also reported a higher prevalence of underweight among lactating women [76]. The factors related to this association are the lack of nutritional education, giving birth at home, first pregnancy below 18 years, and a family size below [76].

## Implications for policy, practice, and future research

These findings highlight critical policy-level concerns. While various government and non-government organisations are actively working to combat undernutrition in Bangladesh, targeted health education programs are needed to address both underweight and overweight issues. For instance, while urban households may benefit from recommendations to reduce fat intake, rural communities need strategies that promote better nutrition to combat underweight prevalence. Public health initiatives must strike a balance to ensure nutritional equity across different regions.

Given the significant rise in obesity in women, community-driven initiatives could be adapted to encourage healthier lifestyles. To combat the rising rates of overweight and obesity, a social and behavioural change communication campaign can be introduced. This initiative should focus on promoting physical activity and raising awareness, particularly among women in their 30s, to prevent weight-related issues. Additionally, future research should examine whether similar trends exist among men and adolescents to achieve a more comprehensive understanding of the issue. Currently, Bangladesh's nutritional programs have limited intervention targeting overweight and obesity [48]. Public education campaigns on physical activity remain slow in implementation [77], and obesity control programs lack sufficient research into implementation barriers. To enhance the effectiveness of interventions, Bangladesh must integrate implementation science and translational research to identify the facilitators and challenges in obesity prevention efforts.

## Strengths and limitations

This study has several strengths. First, it provides up-to-date national estimates on underweight, overweight, and obesity among Bangladeshi women aged between 15–49 years, using a nationally representative dataset. Additionally, the use of standardised questionnaires and calibrated tools enhances the study's validity. However, some limitations must be acknowledged. A key limitation of this study is that it relies on self-reported data, which may introduce biases or inaccuracies. Since the study is cross-sectional, it cannot establish causal relationships between exposures and health outcomes. Additionally, essential variables, such as physical activity and dietary habits, were not collected in the BDHS 2022 dataset, preventing their inclusion in the analysis. Moreover, BMI was used as the primary indicator of nutritional status. However, waist circumference and waist-to-hip ratio—more accurate indicators for cardiovascular disease risk and abdominal obesity—were not assessed. Nonetheless, BMI is the most widely used anthropometric indicator for assessing fatness, and a recent study has shown a significant correlation ($r > 0.8$) with mid-upper arm circumference in underweight Bangladeshi adults [72]. Additionally, Asian cut-offs for BMI were used in this study, which might overestimate obesity rates compared to the WHO criteria. Regular assessment and evaluation of the population's nutritional status are essential to tackle both the persistent issue of undernutrition and the emerging obesity epidemic. Nevertheless, the findings underscore the pressing necessity for intervention to rectify the nutritional disparities affecting women in Bangladesh.

## Conclusion

This study highlights the double burden of malnutrition among Bangladeshi reproductive-aged women using a nationally representative sample. Underweight disproportionately affects younger, poorer, rural, and breastfeeding women, while overweight and obesity are prevalent among older, wealthier, educated, and urban women. These findings call for specific, action-oriented strategies, including the execution of community-based targeted nutrition support programs for young and adolescent women in rural and low-income households to reduce underweight. Promote breastfeeding as a dual strategy for infant health and maternal weight management. Certain physical activity programs can be initiated at the workplace to encourage active lifestyles, particularly for women in sedentary jobs. Policymakers should design subnational interventions for high-risk divisions. Future research can consider a longitudinal design to explore causal pathways between socio-economic changes and weight status; evaluate the integrated interventions addressing both undernutrition and obesity; and explore psychosocial factors, such as mental health or body image, influencing the nutritional status of women.

## Supporting information

**S1 Table. Results of Pseudo-R$^2$, log likelihood, AIC, and BIC to check the goodness of fit of the selected models.** Note: *Model I: fitted without predictor variables (null model); **Model II: fitted with only individual level variables (i.e., women's education, women's employment, household wealth, marital status, women's age, parity, listening to radio, reading magazine, watching television, and currently breastfeeding); ***Model III: fitted with only regional variables (i.e., place of residence, and division); ****Model IV: fitted with all predictor variables (i.e., both individual and regional variables). aAIC: Akaike information criterion; bBIC: Bayesian information criterion.
(PDF)

**S2 Table. Multicollinearity diagnosis result.** Note: aVIF: Variance Inflation Factor.
(PDF)

## Acknowledgments

The authors thank the National Institute of Population Research and Training (NIPORT), Bangladesh, and DHS program for providing the dataset used in this study.

## Author contributions

**Conceptualization:** Ashim Kumar Nandi, Kh Shafiur Rahaman, Navira Chandio, Amit Arora.

**Data curation:** Ashim Kumar Nandi, Kh Shafiur Rahaman, Navira Chandio.

**Formal analysis:** Ashim Kumar Nandi, Kh Shafiur Rahaman, Navira Chandio.

**Methodology:** Ashim Kumar Nandi, Kh Shafiur Rahaman, Navira Chandio, Amit Arora.

**Project administration:** Ashim Kumar Nandi, Kh Shafiur Rahaman, Navira Chandio, Amit Arora.

**Resources:** Ashim Kumar Nandi, Kh Shafiur Rahaman, Navira Chandio, Amit Arora.

**Software:** Ashim Kumar Nandi, Kh Shafiur Rahaman, Navira Chandio.

**Supervision:** Navira Chandio, Amit Arora.

**Validation:** Ashim Kumar Nandi, Kh Shafiur Rahaman, Navira Chandio, Amit Arora.

**Visualization:** Ashim Kumar Nandi, Navira Chandio, Amit Arora.

**Writing – original draft:** Ashim Kumar Nandi, Kh Shafiur Rahaman, Navira Chandio.

**Writing – review & editing:** Ashim Kumar Nandi, Kh Shafiur Rahaman, Navira Chandio, Amit Arora.

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
