## [Decision Letter · Decision Letter 0]

29 Oct 2025

Prevalence and determinants of underweight, overweight, and obesity among reproductive-aged Bangladeshi women: Evidence from Bangladesh Demographic and Health Survey 2022

PLOS ONE

Dear Dr. Nandi,

Thank you for submitting your manuscript to PLOS ONE. After careful consideration, we feel that it has merit but does not fully meet PLOS ONE’s publication criteria as it currently stands. Therefore, we invite you to submit a revised version of the manuscript that addresses the points raised during the review process.

We look forward to receiving your revised manuscript.

Kind regards,

Paul Obeng, MEd, MSc., M.Phil.

Academic Editor

PLOS ONE

Journal Requirements:

Reviewers' comments:

Reviewer's Responses to Questions

**Comments to the Author**

1. Is the manuscript technically sound, and do the data support the conclusions?

Reviewer #1: Partly

Reviewer #2: Yes

2. Has the statistical analysis been performed appropriately and rigorously?

Reviewer #1: Yes

Reviewer #2: No

3. Have the authors made all data underlying the findings in their manuscript fully available?

Reviewer #1: No

Reviewer #2: Yes

4. Is the manuscript presented in an intelligible fashion and written in standard English?

Reviewer #1: Yes

Reviewer #2: Yes

Reviewer #1: The manuscript, in its current form, may contain a critical data error that fundamentally undermines its findings and conclusions. The reported prevalence of obesity (79.08%) is epidemiologically implausible and may indicate a likely miscalculation or misclassification of BMI categories. The study itself cites the BDHS 2017-18, where the combined overweight/obesity rate for women was 45.60%. It is statistically impossible for the obesity rate alone to jump from a fraction of that 45.6% to 79.08% in just 4-5 years without a catastrophic societal shift, which has not occurred. This issue must be resolved before the manuscript can be considered for publication.

Abstract

The abstract selectively highlights "higher education levels" and "greater wealth" as key factors associated with overweight/obesity. However, the full results (Table 3) indicate that the strongest risk factor is older age (ARRR = 1.86 for 35-49 year-olds), followed by watching television (ARRR = 1.21), while the association with higher education is more modest (ARRR = 1.19). The narrative in the abstract should be revised to reflect the magnitude of the associations and prioritize the most influential risk factors identified by the analysis.

The conclusion does not align wholly with the topic. The conclusion should reflect both the prevalence and the determinants. It introduces “the dual burden of malnutrition,” which is relevant, but that concept isn’t explicitly in your title — so it needs to be linked more clearly to your findings.

108 -109: the statement seem to suggest that the prevailing gender inequality in Bangladesh could be attributed to malnutrition in females compared to males. Is that the case?

Introduction

The introduction effectively establishes the public health significance of the double burden of malnutrition and provides a good rationale for focusing on women of reproductive age.

The justification for using the most recent data is well-placed but could be more forcefully articulated by explicitly stating what changes (e.g., economic shifts, urbanization) since the last survey make this update crucial.

Methods and Results

The implausible BMI distribution (79.08% obese, 11.40% normal weight) must be investigated and corrected if an error was made.

The analysis does not account for key behavioural determinants such as dietary patterns and physical activity levels, which are major limitations that should be explicitly acknowledged and discussed.

The entire discussion is based on a possibly incorrect prevalence estimate and must be entirely rewritten once the data error is fixed.

The discussion does not adequately address the limitations of the study, particularly the unmeasured confounding by diet and physical activity, and the cross-sectional nature of the data.

Claims about public health implications are premature until the core findings are validated through data correction.

The conclusion should outline specific, actionable recommendations for public health policy and future research directions.

Reviewer #2: 1. The reported BMI distribution (obesity = 79.08%, normal = 11.40%, overweight = 6.22% in 27,974 women) is implausible for BDHS 2022, even using Asian cut-offs. This likely reflects category misclassification. Please re-check: (i) BMI band recoding, (ii) possible label swap between overweight and obesity, and (iii) weighted percentages and denominators.

2. Present both Asian (23/27.5 kg/m²) and WHO (25/30 kg/m²) cut-off schemes in a sensitivity table or figure and discuss their effect on prevalence.

3. The text says “multilevel multinomial”, but Methods/Software say mlogit with svy, which is single level not multilevel. gsem is multilevel multinomial approach. So revise to “survey-weighted multinomial logistic regression” .

4. In the univariable table, Rural = 0.37 (1.12–1.67) is impossible (CI not bracketing the point). There are several similar formatting/logic errors.

5. The objective should avoid causal wording. Replace “risk factors” with “associated factors,” “determinants,” or “correlates.”

6. The term “significant prevalence” is inappropriate unless a statistical comparison was performed. Use neutral wording such as “a high prevalence” or “a substantial prevalence.”

7. In conclusion the phrase “risk identified” should be revised to “factors associated with” or “determinants identified.”

8. Keep one decimal place consistently throughout tables, text, and figures for presenting percentages.

9. The citation for BMI categorization (references no 31) appears incorrect. Verify that the cited reference truly describes the Asian or Bangladeshi BMI cut-off recommendation (should be: WHO Expert Consultation, Lancet 2004;364:157–63). Check other references accordingly.

**Do you want your identity to be public for this peer review?** For information about this choice, including consent withdrawal, please see our Privacy Policy

Reviewer #1: No

Reviewer #2: No

---

## [Author Response · Author response to Decision Letter 1]

2 Dec 2025

Response to Reviewer 1 Comments

1. Questions for General Evaluation

Is the manuscript technically sound, and do the data support the conclusions?

Reviewer’s Evaluation: Partly

Has the statistical analysis been performed appropriately and rigorously?

Reviewer’s Evaluation: Yes

Have the authors made all data underlying the findings in their manuscript fully available?

Reviewer’s Evaluation: No

Is the manuscript presented in an intelligible fashion and written in standard English?

Reviewer’s Evaluation: Yes

Response and Revisions: Thanks for this evaluation. Please see our point-by-point responses to the comments below.

2. Point-by-point response to Comments and Suggestions for Authors

Comments 1: [The manuscript, in its current form, may contain a critical data error that fundamentally undermines its findings and conclusions. The reported prevalence of obesity (79.08%) is epidemiologically implausible and may indicate a likely miscalculation or misclassification of BMI categories. The study itself cites the BDHS 2017-18, where the combined overweight/obesity rate for women was 45.60%. It is statistically impossible for the obesity rate alone to jump from a fraction of that 45.6% to 79.08% in just 4-5 years without a catastrophic societal shift, which has not occurred. This issue must be resolved before the manuscript can be considered for publication.].

Response 1: Dear Reviewer, thank you for pointing this out. After re-checking the codes, we resolved the coding errors and revised the results when necessary.

Comments 2: [Abstract: The abstract selectively highlights "higher education levels" and "greater wealth" as key factors associated with overweight/obesity. However, the full results (Table 3) indicate that the strongest risk factor is older age (ARRR = 1.86 for 35-49 year-olds), followed by watching television (ARRR = 1.21), while the association with higher education is more modest (ARRR = 1.19). The narrative in the abstract should be revised to reflect the magnitude of the associations and prioritize the most influential risk factors identified by the analysis.

The conclusion does not align wholly with the topic. The conclusion should reflect both the prevalence and the determinants. It introduces “the dual burden of malnutrition,” which is relevant, but that concept isn’t explicitly in your title — so it needs to be linked more clearly to your findings.

108 -109: the statement seem to suggest that the prevailing gender inequality in Bangladesh could be attributed to malnutrition in females compared to males. Is that the case?]

Response 2: Thank you for this feedback. Based on the updated analysis, the entire result and conclusion section was revised and incorporated according to the magnitude of the associations. Conclusion is now liked with the findings.

Comments 3: [Introduction: The introduction effectively establishes the public health significance of the double burden of malnutrition and provides a good rationale for focusing on women of reproductive age.

The justification for using the most recent data is well-placed but could be more forcefully articulated by explicitly stating what changes (e.g., economic shifts, urbanization) since the last survey make this update crucial.]

Response 3: Thank you for your valuable feedback. We have revised the introduction to explicitly state the research gap and the rationale for conducting this study.

Comments 4: [The implausible BMI distribution (79.08% obese, 11.40% normal weight) must be investigated and corrected if an error was made.]

Response 4: Thank you for pointing this out. After re-checking the codes, we resolved the coding errors and revised the results when necessary.

Comments 5: [The analysis does not account for key behavioural determinants such as dietary patterns and physical activity levels, which are major limitations that should be explicitly acknowledged and discussed.]

Response 5: Thank you for the comment. However, we have already addressed the issues under the “Strengths and limitations” subsection in the manuscript.

Comments 6: [The entire discussion is based on a possibly incorrect prevalence estimate and must be entirely rewritten once the data error is fixed.]

Response 6: We have corrected the prevalence estimates and addressed the changes in our discussion section. However, some parts of the discussion were retained as the factors associated with underweight, overweight, and obesity remained similar to what we have found earlier.

Comments 7: [The discussion does not adequately address the limitations of the study, particularly the unmeasured confounding by diet and physical activity, and the cross-sectional nature of the data.

Claims about public health implications are premature until the core findings are validated through data correction.]

Response 7: Thank you for the comment. However, we have already addressed the issues under the “Strengths and limitations” subsection in the manuscript. As mentioned above, we corrected the data and revised the concerned results and discussions accordingly.

Comments 8: [The conclusion should outline specific, actionable recommendations for public health policy and future research directions.]

Response 8: Thank you for your valuable suggestion. We have revised the conclusion to include clear, actionable recommendations for public health policy and outlined future research directions, as requested.

Response to Reviewer 2 Comments

1. Questions for General Evaluation

Is the manuscript technically sound, and do the data support the conclusions?

Reviewer’s Evaluation: Yes

Has the statistical analysis been performed appropriately and rigorously?

Reviewer’s Evaluation: No

Have the authors made all data underlying the findings in their manuscript fully available?

Reviewer’s Evaluation: Yes

Is the manuscript presented in an intelligible fashion and written in standard English?

Reviewer’s Evaluation: Yes

Response and Revisions: Thanks for this evaluation. Please see our point-by-point responses to the comments below.

2. Point-by-point response to Comments and Suggestions for Authors

Comments 1: [1. The reported BMI distribution (obesity = 79.08%, normal = 11.40%, overweight = 6.22% in 27,974 women) is implausible for BDHS 2022, even using Asian cut-offs. This likely reflects category misclassification. Please re-check: (i) BMI band recoding, (ii) possible label swap between overweight and obesity, and (iii) weighted percentages and denominators.].

Response 1: Dear Reviewer, thank you for pointing this out. After re-checking the codes, we resolved the coding errors and revised the results when necessary.

Comments 2: [2. Present both Asian (23/27.5 kg/m²) and WHO (25/30 kg/m²) cut-off schemes in a sensitivity table or figure and discuss their effect on prevalence.]

Response 2: Thank you for pointing this out. We have addressed the comment accordingly.

Comments 3: [3. The text says “multilevel multinomial”, but Methods/Software say mlogit with svy, which is single level not multilevel. gsem is multilevel multinomial approach. So revise to “survey-weighted multinomial logistic regression”.]

Response 3: Thank you for your comment. We revised accordingly.

Comments 4: [4. In the univariable table, Rural = 0.37 (1.12–1.67) is impossible (CI not bracketing the point). There are several similar formatting/logic errors.]

Response 4: Thank you for pointing this out. We agree with your comment. Therefore, we addressed the comment accordingly. We have re-checked other results.

Comments 5: [5. The objective should avoid causal wording. Replace “risk factors” with “associated factors,” “determinants,” or “correlates.”]

Response 5: Thank you for your comment. We revised accordingly.

Comments 6: [6. The term “significant prevalence” is inappropriate unless a statistical comparison was performed. Use neutral wording such as “a high prevalence” or “a substantial prevalence.”]

Response 6: We agree with your comment. The conclusion section is now re-written based on the updated findings.

Comments 7: [7. In conclusion the phrase “risk identified” should be revised to “factors associated with” or “determinants identified.”]

Response 7: The term “risk” was replaced with “associated factors” where possible. The conclusion section was modified.

Comments 8: [8. Keep one decimal place consistently throughout tables, text, and figures for presenting percentages.]

Response 8: Thank you for your comment. We revised accordingly.

Comments 9: [9. The citation for BMI categorization (references no 31) appears incorrect. Verify that the cited reference truly describes the Asian or Bangladeshi BMI cut-off recommendation (should be: WHO Expert Consultation, Lancet 2004;364:157–63). Check other references accordingly.]

Response 9: Thank you for pointing this out. We have updated this reference and added the following:

“WHO Expert Consultation. Appropriate body-mass index for Asian populations and its implications for policy and intervention strategies. Lancet. 2004 Jan 10;363(9403):157-63. doi: 10.1016/S0140-6736(03)15268-3. Erratum in: Lancet. 2004 Mar 13;363(9412):902. PMID: 14726171.”

We have checked other references as well.

Response to Academic Editor comments

Comment 1: [Journal Requirements:

https://journals.plos.org/plosone/s/file?id=wjVg/PLOSOne_formatting_sample_main_body.pdfand
https://journals.plos.org/plosone/s/file?id=ba62/PLOSOne_formatting_sample_title_authors_affiliations.pdf

If the reviewer comments include a recommendation to cite specific previously published works, please review and evaluate these publications to determine whether they are relevant and should be cited. There is no requirement to cite these works unless the editor has indicated otherwise.]

Response 1: Dear Academic Editor, thank you for your comments. We have addressed all the additional requirements accordingly.

---

## [Editor Report · Decision Letter 1]

2 Jan 2026

Prevalence and determinants of underweight, overweight, and obesity among reproductive-aged Bangladeshi women: Evidence from Bangladesh Demographic and Health Survey 2022

PONE-D-25-21645R1

Dear Dr. Nandi,

We are pleased to inform you that your manuscript has been judged scientifically suitable for publication and will be formally accepted for publication once it meets all outstanding technical requirements.

Within one week, you will receive an e-mail detailing the required amendments. When these have been addressed, you will receive a formal acceptance letter, and your manuscript will be scheduled for publication.

If your institution or institutions have a press office, please notify them about your upcoming paper to help maximise its impact. If they will be preparing press materials, please inform our press team as soon as possible -- no later than 48 hours after receiving the formal acceptance. Your manuscript will remain under strict press embargo until 2 pm Eastern Time on the date of publication. For more information, please contact onepress@plos.org.

Kind regards,

Paul Obeng, MEd, MSc, MPhil.

Academic Editor

PLOS One

---

## [Editor Report · Acceptance letter]

PONE-D-25-21645R1

PLOS One

Dear Dr. Nandi,

I'm pleased to inform you that your manuscript has been deemed suitable for publication in PLOS One. Congratulations! Your manuscript is now being handed over to our production team.

Kind regards,

on behalf of

Dr. Paul Obeng

Academic Editor

PLOS One